# POLICYRAG: PROMPT-GUIDED SYMBOLIC GRAPH MEMORY FOR INTERPRETABLE MULTI-HOP RE-TRIEVAL

## ABSTRACT

Retrieval augmented generation is a powerful way to ground large language models in external knowledge, yet most pipelines still treat the prompt as instructions for text production rather than as a control surface for retrieval. We introduce PolicyRAG, a framework that recasts retrieval as an explicit, auditable policy operating over a symbolic graph memory. Text is organized into lightweight typed links between entities and passages, enabling transparent search and controllable evidence selection. Beyond accuracy, the policy is compact and human editable, supporting governance, domain adaptation, and safety review without retraining. At query time, the policy seeds candidate entities, invokes brief LLM calls only for disambiguation and local gating, and performs symbolic traversal with Personalized PageRank (PPR). The resulting scores are projected to passages and finalized with a small, transparent re-ranker, producing a per-query trace of seeds, paths, and scores for explainable evidence selection. Compared with long-context expansion, the policy keeps test-time compute modest while preserving answer quality. On multi-hop question answering benchmarks, PolicyRAG achieves state-of-the-art results on HotpotQA (F1 80.7), 2WikiMultiHopQA (F1 78.9), and MuSiQue (F1 55.9) while remaining fully auditable and training free. We also assess domain adaptability on domain-specific datasets. By coupling symbolic structure with prompt level control, PolicyRAG provides a practical route from question to verifiable evidence and advances accuracy, efficiency, and trust in retrieval augmented generation. All source code and project files are available at this link[1].

## 1 INTRODUCTION

Large language models (LLMs) demonstrate remarkable capabilities in text completion, summarization, and reasoning tasks, yet exhibit fundamental limitations when required to synthesize dynamic knowledge from multiple sources with verifiable evidence chains. While retrieval-augmented generation (RAG) frameworks address knowledge staleness by integrating external data sources, conventional dense passage retrieval techniques based on semantic similarity exhibit systematic failures in multi-hop reasoning scenarios where evidence fragments are distributed across documents and connected solely through latent entity relationships (Karpukhin et al., 2020; Lewis et al., 2020b). Performance disparities on MultiHop QA benchmarks –particularly HotpotQA (Yang et al., 2018), 2WikiMultiHopQA (Ho et al., 2020), and MuSiQue (Trivedi et al., 2022)—reveal that embedding-based retrieval inadequately captures the compositional reasoning structures necessary for evidence synthesis.

**The Retrieval Policy Problem:** Consider the multi-hop question: *"When was the person who Messi's goals in Copa del Rey were compared to signed by Barcelona?"* This query necessitates a three-stage reasoning chain: (1) identifying that Messi's goals were compared to Diego Maradona's "Goal of the Century," (2) establishing that Maradona was signed by Barcelona, and (3) retrieving the specific signing date (June 1982). Traditional dense retrieval systems fail systematically on such queries because:

---

[1] https://mega.nz/folder/nlpBgKBT#tEVS7O_wPI-DSis-3UDHmg

- **Semantic similarity alone is insufficient** to traverse the comparison relationship (*compared to*) that bridges Messi and Maradona

- **Distractor documents** containing mentions of "Messi," "Barcelona," and "Copa del Rey" score highly on embedding similarity but provide no information about Maradona

- **The reasoning path** (*Messi → compared to → Maradona → signed by → Barcelona → date*) requires explicit typed relations rather than latent vector space proximity

Existing graph-based approaches partially address these limitations by materializing entity relationships, yet retain fundamental constraints: they primarily deploy prompts as generative mechanisms rather than retrieval control surfaces, thereby forfeiting opportunities for dynamic evidence filtering, query-specific graph navigation policies, and transparent audit trails (Asai et al., 2020; Sun et al., 2018). Post-retrieval stages in contemporary architectures prioritize generation-oriented prompting while neglecting the capacity of language models to provide targeted retrieval supervision through structured policy mechanisms. This architectural decision precludes adaptation to domain-specific requirements, evidence quality constraints, and governance mandates without extensive retraining.

**Policy-Driven Retrieval Architecture:** We propose a paradigmatic shift from prompt-as-generator to prompt-as-policy, wherein each query instantiates an explicit, auditable retrieval controller operating over a symbolic graph memory. The PolicyRAG framework conceptualizes retrieval as a structured policy governing: (1) entity seed selection through relevance-weighted restart distributions, (2) local fact retention via keep-drop gating prompts that filter irrelevant triples, (3) typed graph traversal using Personalized PageRank (PPR) with relation-specific edge weighting, (4) multi-factor passage scoring with length and specificity normalization, and (5) transparent re-ranking emphasizing title concordance and multi-seed coverage. This architecture maintains complete decision audit trails— seeds, filtered facts, traversal paths, and ranking scores—enabling governance review, domain adaptation, and iterative policy refinement without model retraining.

In the above example, the policy controller extracts entity seeds {*Messi*, *Barcelona*, *Copa del Rey*}, identifies candidate facts near these seeds, applies keep-drop filtering to retain only comparison and signing relationships, executes typed PPR with elevated weights on RELATED edges versus SIMILAR_TO edges, projects entity scores to passages via mention-based incidence, and re-ranks based on multi-seed coverage. The resulting top-5 passages contain the complete reasoning chain with gold evidence ranked first, demonstrating how explicit policy control surfaces relevant information that embedding similarity alone obscures.

**Empirical Validation and Theoretical Contributions:** We evaluate PolicyRAG on three established multi-hop QA benchmarks under controlled experimental protocols with matched computational budgets. The framework achieves state-of-the-art performance: F1 80.7% on HotpotQA, 78.9% on 2WikiMultiHopQA, and 55.9% on MuSiQue, while maintaining superior early-rank recall (Recall@5: 97.1, 98.1, 78.9 respectively) and complete auditability. Critically, these results are obtained without any task-specific training, through purely symbolic graph operations and compact policy specifications. Ablation studies isolate the contribution of typed connectivity, local fact gating, and normalization strategies, revealing that explicit policy control over evidence paths –rather than expanded context windows or learnt re–rankers -drives performance gains.

This work advances retrieval-augmented generation along three dimensions:

1. **Architectural Innovation:** We reframe the prompt as a retrieval policy rather than a generation instruction, establishing a training-free framework for policy-driven evidence selection over symbolic graph memory with typed connectivity.

2. **Methodological Rigor:** We demonstrate that symbolic memory controllers can achieve effective retrieval control through five key mechanisms—entity extraction with relevance ranking, keep-drop prompting for fact filtering, Personalized PageRank traversal, multi-factor passage scoring, and structured re-ranking—while maintaining complete decision audit trails for transparency and verification.

3. **Empirical Validation:** We report state-of-the-art performance on multi-hop question answering benchmarks while preserving interpretability, auditability, and domain transferability, validated through systematic ablation studies and domain-specific corpus evaluation.

By coupling symbolic structure with prompt-level control, PolicyRAG provides a practical route from question to verifiable evidence, advancing accuracy, efficiency, and trust in retrieval-augmented generation systems.

## 2 RELATED WORKS

The development of retrieval-augmented generation systems for multi-hop reasoning requires integrating insights from three complementary domains: (1) graph-enhanced architectures materializing entity relationships, (2) neurobiologically-inspired memory systems modeling associative recall, and (3) policy-driven retrieval frameworks enabling interpretable evidence selection. This organization identifies architectural innovations, methodological limitations, and synthesis opportunities motivating PolicyRAG's design.

### 2.1 MULTI-HOP QA WITH LLMS AND GRAPHS

Multi-hop question answering presents fundamental challenges when crucial information is distributed across documents linked solely by latent entity associations. Large language models excel at single-step reasoning but exhibit diminished effectiveness when solutions require integrating information from multiple sources with traceable evidence chains (Yang et al., 2018; Ho et al., 2020; Trivedi et al., 2022). Conventional retrieval-augmented generation frameworks struggle with multi-hop queries, as embedding similarity inadequately captures compositional associations essential for multi-step reasoning (Karpukhin et al., 2020; Lewis et al., 2020b).

**Graph-Enhanced Architectures:** Graph-structured knowledge representations enable discovery of entity relationships and passage connections, enhancing retrieval efficacy. QA-GNN established methodologies for integrating textual reasoning with structured knowledge graphs through dual-component architectures (Yasunaga et al., 2021), while GreaseLM advanced state-of-the-art performance via innovative text-graph encoding integration (Zhang et al., 2022). Think-on-Graph enables large language models to navigate knowledge graphs without retraining (Sun et al., 2024). Contemporary structure-augmented approaches include RAPTOR, employing recursive clustering for hierarchical text organization with 20% QuALITY benchmark gains (Sarthi et al., 2024), GraphRAG's elaborate knowledge graphs for comprehensive reasoning (Edge et al., 2024), and LightRAG's computationally efficient graph-based retrieval (Guo et al., 2024).

*Advancement Towards Biologically-Inspired Computational Architectures:* Graph-based frameworks are typically utilized to represent the relationships between entities by creating static structures, which are then navigated using predefined algorithms to retrieve information. However, a novel paradigm, inspired by the associative recall mechanisms of the hippocampus, suggests that memory operations that are dynamic and sensitive to the context in which they operate provide a more efficient framework for conducting multi-hop reasoning tasks.

**Systems Inspired by Neurobiological Processes:** Computing architectures that incorporate principles of hippocampal memory have demonstrated notable gains in efficiency: for instance, the HippoRAG model achieved a 20% enhancement in performance while simultaneously reducing computational demands by a factor of 10 to 30. This was accomplished by employing the theory of hippocampal indexing (Gutiérrez et al., 2024). Furthermore, the subsequent iteration, HippoRAG2, was able to achieve an additional improvement of 7% (Gutiérrez et al., 2025). The efficacy of integrating neural-symbolic methods was further substantiated by the Generate-on-Graph and Graph Chain-of-Thought approaches (Xu et al., 2024; Jin et al., 2024).

*Integrative Analysis:* Despite the capacity of graph architectures to represent relationships between entities and the demonstrated efficiency of neurobiologically inspired systems in associative memory recall, these frameworks predominantly utilize prompts as generative tools rather than mechanisms for explicit retrieval control. This limitation results in an underutilization of language models' full potential for precise retrieval supervision, dynamic filtering of evidence, navigation tailored to specific queries, meticulous control over traversal processes, evaluation of evidence quality, and monitoring of provenance. Consequently, these systems face challenges in adapting to the specific requirements of various domains without necessitating retraining.

*Development of Policy-Oriented Frameworks:* A promising new research direction is reimagining prompts as explicit retrieval policies that guide the selection of evidence, the process of graph traversal, and the implementation of scoring mechanisms. This paradigm shift facilitates systematic retrieval control through the use of symbolic memory frameworks, thereby enhancing interpretability and auditability while retaining the flexibility inherent in natural language specifications. By doing so, it allows for a more structured approach to managing and utilizing information retrieval processes.

## 3 POLICYRAG FRAMEWORK

PolicyRAG views the prompt policy as an explicit retrieval policy operating on a symbolic graph memory. In this section, we describe the pipeline components: symbolic memory construction using LLM-driven entity and triple extraction, and query-time policy execution consisting of seeding, fact filtering, typed PPR traversal, passage scoring, and answer synthesis, as shown in Figure 1. All design choices are made explicit, and the exact prompts policy used in our implementation is shown. An example of PolicyRAG for multi-hop QA is shown in Figure 2

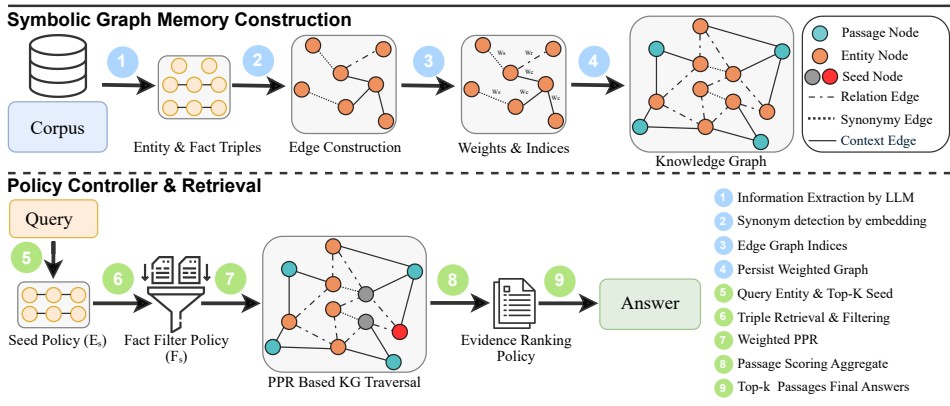

Figure 1: **PolicyRAG methodology.** Steps include symbolic graph memory construction through entity and fact triple extraction, weighted graph persistence, knowledge graph construction, seed selection and fact filtering, traversal via Personalized PageRank (PPR), passage scoring and reranking, and final answer synthesis.

**Notation.** We collect notation here and provide a detailed symbol glossary in Appendix F. In short:

$$\mathcal{G} = (V, \mathbb{E}) \quad \text{with} \quad V = V_E \cup V_P, \ V_E \cap V_P = \varnothing,$$

where $V_E$ are the entity nodes and $V_P$ are the passage nodes. Edges are partitioned into context edges $E_c$, typed relation edges $E_r$, and synonymy edges $E_s$. The Row-stochastic adjacency operators for these families are denoted $A_c, A_r, A_s$. $B \in \{0,1\}^{|V_E| \times |V_P|}$ is the entity–passage incidence matrix. For any integer $n$, $\Delta^n$ denotes the $n$-dimensional probability simplex.

### 3.1 SYMBOLIC GRAPH MEMORY CONSTRUCTION

The symbolic graph is constructed once per corpus. We outline each step and include the prompts used in our implementation.

**Entity extraction via fixed LLM prompt.** For each passage $p$ we extract a canonicalized list of named entities $E(p)$. The extraction is performed with a high-precision LLM prompt:

```
ENTITY EXTRACTION PROMPT POLICY: Extract PERSON, ORG,
LOCATION, DATE, EVENT, PRODUCT...
Normalize duplicates. Output strict JSON: {"named_entities":
[...]}.
```

We retain entities whose type is non-generic (e.g., PERSON, ORG, GPE, DATE, EVENT). To measure entity specificity, we use inverse-passage frequency:

$$\text{spec}(e) \;=\; \frac{1}{1 + \text{df}(e)}, \qquad \text{df}(e) := \#\{p : e \in E(p)\}. \tag{1}$$

**Triple extraction using constrained RDF prompt.**   Given entities for a passage, we extract candidate factual triples (subject, predicate, object) using a constrained prompt:

```
TRIPLE EXTRACTION PROMPT POLICY: Construct RDF triples
[subj, pred, obj],
using only entities from the named_entities list.  Output
strict JSON: {"triples":[...]}.
```

We keep triples whose subject and object are in-vocabulary, whose predicates belong to a compact and non-generic predicate set, and whose statements are non-conflicting (with conflicts resolved via confidence or majority voting). The resulting typed relation predicates define the edge family $E_r$.

**Context and synonymy edges.**   For each entity $e$ mentioned in the passage $p$, we add the context edge $(e, p) \in E_c$. Optional synonym edges $E_s$ are added either from LLM-detected abbreviations (e.g., RFEF $\leftrightarrow$ Royal Spanish Football Federation) or by high-threshold embedding similarity.

**Deduplication and persistence.**   Duplicate triples with identical $(s, p, o)$ are merged; conflicting triples are resolved by majority filtering. We persist with sparse adjacency operators $A_c, A_r, A_s$, entity specificity $\text{spec}(e)$, passage lengths $\ell(p)$, the incidence matrix $B$, and nonnegative edge-type weights $(\eta_c, \eta_r, \eta_s)$ used at query time.

**Composite transitions.**   Define composite transitions that operate on the entity layer and between entity and passage layers:

$$T_E \;=\; \eta_r A_r + \eta_s A_s, \qquad T_{E \to P} \;=\; \text{row-norm}(B), \qquad T_{P \to E} \;=\; B^\top. \tag{2}$$

$T_E$ navigates the entity layer (typed relations + alias correction). $T_{E \to P}$ transfers the mass of the entity to passages through context mentions.

## 3.2   QUERY–TIME POLICY CONTROLLER

At inference time, a compact, training-free controller governs seed selection, fact filtering, typed traversal, and answer generation. In the following, we describe each component.

**Seed scoring policy.**   Given a query $q$ we compute a restart (seed) distribution $s \in \Delta^{|V_E|}$ over entity nodes:

$$s_e \;=\; \frac{\text{align}(e, q)\, \text{spec}(e)\, \text{aliasOK}(e)}{\sum_{e'} \text{align}(e', q)\, \text{spec}(e')\, \text{aliasOK}(e')}. \tag{3}$$

Here $\text{align}(e, q)$ is a (possibly combined) lexical and embedding similarity between entity $e$ and query $q$, and $\text{aliasOK}(e) \in \{0, 1\}$ is an indicator that filters out overly broad aliases.

**Fact gating using an explicit filtering prompt.**   Let $E_s$ denote a (small) set of high-probability seed entities sampled from $s$ (e.g., top-$k$ mass or mass threshold). We gather adjacent candidate facts:

$$\text{Cand}(E_s) \;=\; \{f : f \text{ is a triple adjacent to some } e \in E_s\}.$$

Filtered facts $F_s$ are selected according to the following prompt policy:

```
FACTS TRIPLES FILTER PROMPT POLICY: Select up to K facts
needed to answer the query.
Policy:  direct > contextual, minimal bridges, specific >
generic.
Output strict JSON: {"fact":  [[s,p,o],...]}.  No new facts
allowed.
```

This prevents distractor-heavy drift in multi-hop retrieval.

**Typed Personalized PageRank (PPR).** We run PPR over the entity layer using the transition $T_E$ from equation 2:

$$v^{(t+1)} = (1-\alpha)\, T_E^\top v^{(t)} + \alpha s, \qquad v^{(0)} = s, \tag{4}$$

where $\alpha \in (0,1]$ is the probability of restart. The iteration continues until convergence; denote the fixed point by $v^\star$.

**Projection to passages and normalization.** Map entity scores to passage scores via the incidence matrix:

$$u = B^\top v^\star \in \mathbb{R}^{|V_P|}.$$

Normalize by passage length and a passage-level inverse-passage-frequency:

$$\tilde{u}_p = \frac{u_p}{\ell(p)^\beta\, \mathrm{IPF}(p)^\gamma}, \qquad \beta, \gamma \in [0,1], \tag{5}$$

where $\ell(p)$ is the length of the passage (e.g. tokens) and $\mathrm{IPF}(p)$ is a passage-specific penalty (see the glossary for the definition in Appendix F).

**Re-ranking.** A final re-ranking score combines normalized mass and several heuristic/learned features:

$$S(p \mid q) = \tilde{u}_p + \lambda_{\text{title}}\, \phi_{\text{title}}(p,q) + \lambda_{\text{cov}}\, \phi_{\text{cov}}(p, E_s) + \lambda_{\text{path}}\, \phi_{\text{path}}(p). \tag{6}$$

Each $\phi$ is a real-valued feature (higher is better) and the $\lambda$'s are nonnegative weights.

**Answer synthesis with strict grounding.** Finally, an LLM is prompted to produce an answer grounded in the retrieved passages and the retained structured facts:

```
ANSWER SYNTHESIS PROMPT POLICY: Answer using only retrieved
passages;
otherwise return ``Not found in retrieved context''.
```

This constraint enforces traceable grounding and reduces hallucinations.

## 4 EXPERIMENTAL SETUP

### 4.1 BASELINES

We compare against three complementary families under a single controlled protocol that fixes the passage inventory, data set splits, and inference budget. First, to anchor the results in established retrieval practice, we include the BM25 lexical and dense retrievers for sparse term matching alongside widely used dense encoders such as DPR, Contriever, and GTR, and a standard RAG pipeline that couples dense retrieval with generation (Robertson & Walker, 1994; Lewis et al., 2020a; Izacard et al., 2021; Ni et al., 2021). Second, to reflect the modern retrieval capacity independent of the graph structure, we add large embedding models that report strong BEIR performance, including GTE-Qwen2-7B-Instruct, GritLM-7B and NV-Embed-v2 (Thakur et al., 2021; Li et al., 2023; Muennighoff et al., 2024; Lee et al., 2024). Third, to test whether explicit connectivity improves evidence finding and answerability under the same compute envelope, we evaluate the structure-augmented RAG methods RAPTOR, which organises the corpus hierarchically, GraphRAG and LightRAG, which exploit the graph structure for retrieval and summarization, and HippoRAG and HippoRAG 2, which integrate a knowledge graph with diffusion-style scoring (Sarthi et al., 2024; Edge et al., 2024; Guo et al., 2024; Jimenez Gutierrez et al., 2024; Gutiérrez et al., 2025). All baselines operate on the identical passage pool with matched token and latency budgets when public checkpoints and recommended hyperparameters are available; otherwise, we retain the reported configurations, with full settings provided in supplementary material in Appendix F. To disentangle retrieval quality from context length, we also report a long-context control that concatenates the same retrieved passages to the generator without any graph reasoning.

## 4.2 DATASETS

We evaluated three established multi-hop question-answering benchmarks that probe complementary aspects of complex reasoning, each constructed from or aligned with Wikipedia[2]. This choice ensures a common provenance while stressing different retrieval and composition behaviors that matter for graph-enhanced RAG. HotpotQA (Yang et al., 2018) contains questions that require evidence drawn from multiple articles, prominently including bridge and comparison types. Each question is paired with gold supporting sentences, enabling faithful evaluation of multi-document reasoning and evidence use. 2WikiMultiHopQA (Ho et al., 2020) is built from curated pairs of Wikipedia pages and emphasizes clean two-hop chains across entities and pages its construction reduces lexical shortcuts and encourages genuine cross-article inference.

Table 1: Dataset statistics

|  | NQ | PopQA | 2Wiki | HotpotQA | MuSiQue |
|---|---|---|---|---|---|
| Number of Queries | 1,000 | 1,000 | 1,000 | 1,000 | 1,000 |
| Number of Passages | 9,633 | 8,676 | 6,119 | 9,811 | 11,656 |

MuSiQue (Trivedi et al., 2022) composes questions from single-hop sub-questions but injects strong distractors, stressing keep drop decisions and robustness to off-topic yet topically related content. For completeness, Table 1 also lists corpus sizes for NQ (Wang et al., 2024) and PopQA (Mallen et al., 2022), which are commonly used for single-hop factual QA. We include them to standardize corpus statistics, our primary multi-hop evaluation focuses on HotpotQA, 2Wiki, and MuSiQue.

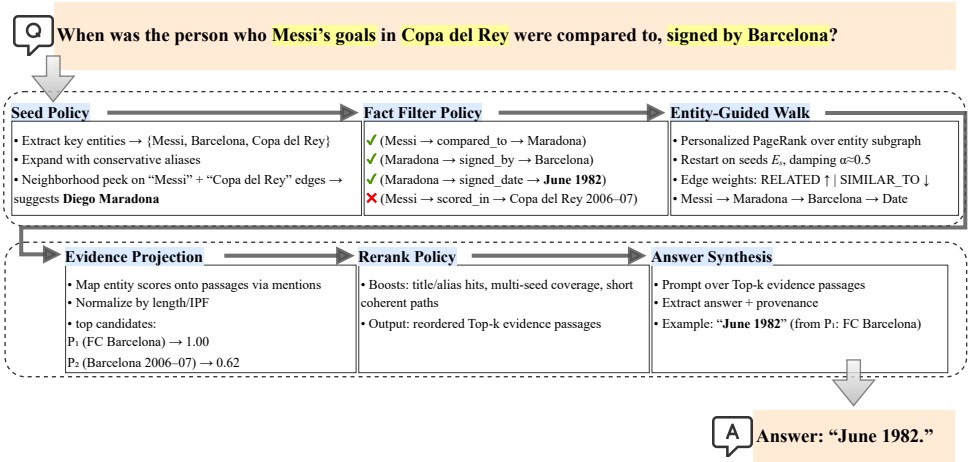

Figure 2: Example of the Policy-Driven Retrieval-Augmented Generation (RAG) framework for multi-hop question answering.

**Domain-specific corpora:** To assess domain adaptability without retraining, we additionally evaluate domain datasets in Legal, Health, and History details in supplementary material in Appendix C. These corpora provide out-of-distribution terminology, citation styles, and entity linking patterns, enabling a targeted stress test of policy editability, governance, and transfer.

## 4.3 EVALUATION METRICS

Our primary answer metric is token-level F1 on normalized strings, following prior multi-hop QA work(Yang et al., 2018; Ho et al., 2020; Trivedi et al., 2022), and we also report exact match (EM) for completeness. Retrieval quality is measured by recall@$k$ with an emphasis on early ranks, and is complemented by MRR and nDCG to capture the quality of ordering. Beyond these task metrics, we track diagnostics that assess whether the controller is selecting evidence that actually supports the answer rather than simply accumulating more text answer-supported@$k$ is the fraction of questions

---

[2]https://www.wikipedia.org/

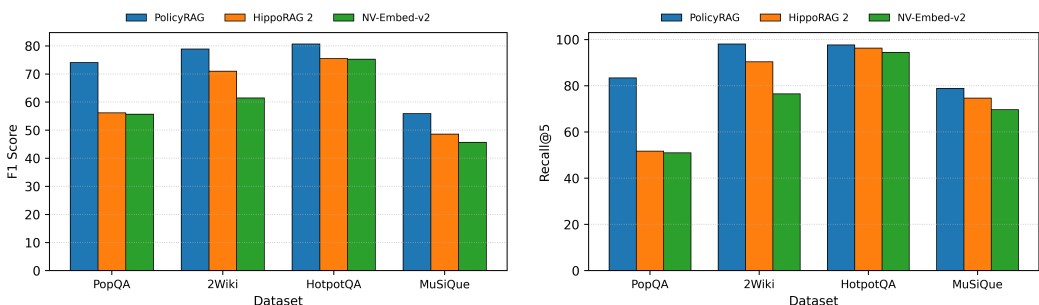

Figure 3: PolicyRAG performance: F1 Score and Recall@5 across PopQA, 2Wiki, HotpotQA, and MuSiQue.

Table 2: QA performance (F1 scores) on RAG benchmarks using GPT-4o-mini as the QA reader. All structure-augmented RAG baselines and PolicyRAG use GPT-4o-mini for structure generation and NV-Embed-v2 for retrieval, while PolicyRAG* uses GPT-4.1-mini. Bold values indicate the best performance in each column.

| Method | Simple QA | | Multi-hop QA | | | Avg |
|---|---|---|---|---|---|---|
| | NQ | PopQA | 2Wiki | HotpotQA | MuSiQue | |
| *Simple Baselines* | | | | | | |
| NV-Embed-v2 (7B) (Lee et al., 2024) | 59.9 | 55.8 | 60.8 | 71.0 | 46.0 | 58.7 |
| *Structure-Augmented RAG* | | | | | | |
| RAPTOR (Sarthi et al., 2024) | 54.5 | 55.1 | 48.4 | 64.7 | 39.2 | 52.3 |
| GraphRAG (Edge et al., 2024) | 55.5 | 51.3 | 61.0 | 67.6 | 42.0 | 55.4 |
| LightRAG (Guo et al., 2024) | 15.4 | 14.8 | 12.1 | 20.2 | 9.3 | 14.3 |
| HippoRAG (Jimenez Gutierrez et al., 2024) | 52.2 | 56.2 | 67.3 | 60.0 | 35.9 | 54.3 |
| HippoRAG 2 (Gutiérrez et al., 2025) | 60.0 | 55.7 | 69.7 | 71.1 | 49.3 | 61.1 |
| **PolicyRAG** | **66.2** | **72.6** | **77.3** | **78.9** | **54.6** | **69.9** |
| **PolicyRAG*** | **68.1** | **74.1** | **78.9** | **80.7** | **55.9** | **71.5** |

whose gold answer is derivable from the top-$k$ passages without external context HopsRecovered@$k$ measures how often the entity chain implied by the gold rationale is fully discoverable within the top-$k$ and the Distractor Suppression Rate quantifies the proportion of retrieved items that are subsequently rejected by the local fact gate near the seeds. To assess robustness, we also measure the stability of ranked lists under small paraphrases and minor alias substitutions in the query. Taken together, these metrics expose precision coverage trade-offs and the specific contribution of typed traversal and gating, complementing headline F1 and EM and aligning with recent evaluations of structure-augmented RAG.

## 4.4 IMPLEMENTATION DETAILS

PolicyRAG is entirely training-free and requires minimal configuration across all datasets. We construct the symbolic memory once per corpus (Wikipedia-aligned) and use it as a read-only structure at inference time. At query time, the controller executes a lightweight seed policy, consisting of a concise keep–drop prompt for local fact gating and a typed PPR traversal with fixed damping and type-mixing parameters. For information extraction, we employ both GPT-4o-mini and GPT-4.1-mini within our framework, allocating each model to complementary roles in the retrieval and reasoning pipeline. Entity-aligned candidate facts are retrieved and ranked using a unified NV-Embed-v2 embedding model, from which the top-5 triples are selected for policy filtering. The QA module conditions strictly on the top-5 retrieved passages using an evidence-first prompt and deterministic decoding. Entity relevance scores are softly projected onto passages using length-aware normalization and combined with an interpretable additive reranking scheme. All fixed hyperparameters including traversal settings, thresholds, and runtime weights are documented in Appendix F.

Table 3: Retrieval performance (passage recall@5) on simple QA and multi-hop QA datasets. The compared structure-augmented RAG methods are reproduced with the same LLM and retriever as ours for a fair comparison. GraphRAG and LightRAG are not presented because they do not directly produce passage retrieval results.

| Method | Simple QA | | Multi-hop QA | | | Avg |
|---|---|---|---|---|---|---|
| | NQ | PopQA | 2Wiki | HotpotQA | MuSiQue | |
| *Simple Baselines* | | | | | | |
| Contriever (Izacard et al., 2021) | 54.6 | 43.2 | 57.5 | 75.3 | 46.6 | 55.4 |
| GTR (T5-base) (Muennighoff et al., 2024) | 63.4 | 49.4 | 67.9 | 73.9 | 49.1 | 60.7 |
| *Large Embedding Models* | | | | | | |
| GritLM-7B (Ni et al., 2021) | 76.6 | 50.1 | 76.0 | 92.4 | 65.9 | 72.2 |
| NV-Embed-v2 (7B) (Lee et al., 2024) | 75.4 | 51.0 | 76.5 | 94.5 | 69.7 | 73.4 |
| *Structure-Augmented RAG* | | | | | | |
| RAPTOR (Sarthi et al., 2024) | 69.4 | 48.1 | 66.0 | 90.2 | 61.0 | 66.9 |
| HippoRAG (Jimenez Gutierrez et al., 2024) | 45.1 | 52.2 | 87.0 | 78.5 | 52.4 | 63.0 |
| HippoRAG 2 (Gutiérrez et al., 2025) | 76.4 | 52.2 | 90.2 | 95.7 | 74.2 | 77.7 |
| **PolicyRAG** | **80.6** | **81.2** | **97.1** | **96.8** | **76.6** | **86.4** |
| **PolicyRAG**$^*$ | **82.0** | **83.4** | **98.1** | **97.1** | **78.9** | **87.9** |

## 5 RESULTS AND DISCUSSIONS

We evaluate our approach on three established multi-hop QA benchmarks HotpotQA, 2Wiki, and MuSiQue using GPT-4o-mini as the primary decoding engine. For comparison, we also report results using GPT-4.1-mini, while keeping the retrieval pipeline identical across all settings. All models are conditioned strictly on the evidence retrieved by our policy-driven graph controller (Yang et al., 2018; Ho et al., 2020; Trivedi et al., 2022). Answer quality is evaluated using token-level F1 and EM, and retrieval quality is assessed using recall@$k$, MRR, and nDCG, together with diagnostic measures of evidence sufficiency and robustness. Summary results are provided in Table 2 (QA performance) and Table 3 (retrieval performance).

**Overall performance:** PolicyRAG attains strong accuracy across all benchmarks 80.7 F1 on HotpotQA, 78.9 F1 on 2Wiki, and 55.9 F1 on MuSiQue while remaining training-free and fully auditable. Retrieval is both early and precise recall@5 reaches 97.1 on HotpotQA, 98.1 on 2Wiki, and 78.9 on MuSiQue, indicating that gold evidence is surfaced with a small candidate budget. On MuSiQue's distractor-heavy setting, the keep–drop controller maintains competitive F1 and preserves early-rank coverage, suggesting the policy's bias toward compact, high-yield evidence sets rather than broader context stuffing. We also validate transfer on domain-specific corpora (Legal, Health, History) without retraining details appear in supplementary material in Appendix C. Figure 3 illustrates these performance trends across datasets.

**Ranking behaviour and coverage:** Ranked lists concentrate verifiable evidence near the top. MRR and nDCG consistently exceed dense-retriever and reader-only controls, reflecting more stable ordering under matched compute. On 2Wiki, HopsRecovered@$k$ saturates quickly, consistent with efficient discovery of both ends of the two-hop chain HotpotQA shows a similar pattern on bridge questions. answer-supported@$k$ tracks recall@$k$ closely, implying that answers are typically derivable from retrieved passages without relying on ungrounded model priors.

**Ablation study:** We ablate the controller along four axes seeding, gating, connectivity, and scoring to isolate which decisions drive end-to-end gains. Removing seed policy specificity yields noisier traversals and weaker early-rank recall. Disabling local fact gating expands the candidate set but depresses answer-supported@$k$ and harms MRR and nDCG, indicating that compact neighborhoods, not larger ones, improve answerability detailed outcomes appear in Table 4. Dropping synonymy edges reduces robustness to aliasing, while removing typed relation edges weakens multi-hop discovery in both cases, early-rank coverage declines even when recall at larger $k$ is similar, underscoring the role of typed connectivity in arriving early. Turning off length and specificity normalization biases projection toward long or frequent passages and degrades ordering quality, and omitting the transparent re-ranking flattens the head of the list by underweighting title and alias hits as well as

multi-seed coverage. Adjusting the PPR damping factor exhibits the expected precision exploration trade-off lower damping explores more but diffuses into off-path regions, whereas higher damping is conservative but risks under-exploration.

Table 4: Ablation study: effect of graph design choices on passage recall@5 in multi-hop QA.

| Method | 2Wiki | HotpotQA | MuSiQue | Avg |
|---|---|---|---|---|
| PolicyRAG[*] | 98.1 | 97.1 | 78.9 | 91.3 |
| w/ NER to node | 98.9 | 79.6 | 58.0 | 78.8 |
| w/ Query to node | 73.2 | 69.1 | 49.1 | 63.8 |
| w/o Passage Node | 98.0 | 89.7 | 67.9 | 85.2 |
| w/o Filter | 98.4 | 96.2 | 77.2 | 90.6 |

**Efficiency and interpretability:** At matched token and latency budgets, the controller processes a similar number of passages per query as baselines but surfaces fewer irrelevant items, yielding smaller, easier-to-audit evidence sets. Per-query traces seeds, kept facts, paths, and scores make the retrieval path auditable end to end and immediately actionable for policy edits, enabling governance without retraining while sidestepping the overheads of long-context expansion. Overall, the findings show that a compact policy over a symbolic graph delivers accurate, efficient, and interpretable multi-hop reasoning, and remains robust in distractor-heavy settings.

## 6 CONCLUSION

PolicyRAG reframes the prompt as a compact retrieval policy over a symbolic graph, pairing targeted seeding, precise fact gating, and typed traversal to surface verifiable evidence sets. Symbolic graphs are vulnerable to incompleteness and noise, where missing entities or incorrect relations can degrade retrieval quality; we address this through targeted seeding that focuses on high-confidence entity matches and a gating mechanism that filters low-relevance facts before traversal. Additionally, fixed traversal depth may miss evidence requiring longer relational chains, though our typed edge traversal partially mitigates this by prioritizing semantically relevant paths within the bounded neighborhood. Building on these results, we are currently investigating path-based retrieval policies that traverse longer relational chains while preserving efficiency and per-query traceability. Our near-term focus is to tighten the coupling between gating and traversal to further lift F1. We are also broadening evaluation to large, domain-specific corpora, moving beyond preliminary probes toward rigorously designed, systematic studies. Together, these efforts will advance PolicyRAG toward a path-level retrieval paradigm that preserves accuracy at scale while remaining editable, auditable, and practical.

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

APPENDIX

This appendix provides an in-depth overview of the components underlying our approach, with detailed elaboration on the following key aspects:

- Appendix A: LLM Prompts Policy

- Appendix B: PolicyRAG Pipeline walk-through

- Appendix C: Detailed Experimental Results

- Appendix D: Error Analysis

- Appendix E: Graph Statistics

- Appendix F: Implementation Details and Hyperparameters

## A    LLM PROMPT POLICIES FOR POLICYRAG

We provide an example demonstration of the fact filtering prompt policy in Figure 4. Alongside this, we define harmonized prompt policies for entity extraction, triple construction, seed selection, domain adaptation, and answer generation together, these policies deliver strong and consistent results across our evaluations.

## B    POLICYRAG PIPELINE WALK-THROUGH

Table 5 shows the end-to-end online loop parsing a question into candidate entities and local facts, keep–drop gating to remove distractors, typed PPR over the entity graph, projection to passages with light normalization, transparent re-ranking, top-$k$ evidence selection, and an emitted per-query trace for audit and reuse.

Table 5: PolicyRAG query processing and response generation. Compact view of entities, facts, and reasoning steps leading to the final answer.

| Original Query | What county contains the city with a radio station that broadcasts to the capital city of the state where the Peace Center is located? |
|---|---|
| Gold Answer | Richland County |
| Entities | Peace Center; Greenville; South Carolina; WWNQ; Forest Acres; WFFG-FM; Warren County |
| Facts (Triples) | ( Peace Center, located in, Greenville ) 
 ( Greenville, in, South Carolina ) 
 ( WWNQ, licensed to, Forest Acres ) 
 ( WFFG-FM, located in, Warren County ) |
| Retrieved Info | Forest Acres is in Richland County, SC; WWNQ serves Columbia (capital of SC); Peace Center is in Greenville, SC |
| Response | Forest Acres is a city in Richland County, South Carolina, United States. |
| Reasoning Chain | Peace Center $\rightarrow$ Greenville $\rightarrow$ South Carolina $\rightarrow$ Columbia (capital) $\rightarrow$ WWNQ $\rightarrow$ Forest Acres $\rightarrow$ Richland County |

## C    DETAILED EXPERIMENTAL RESULTS

We report full QA and retrieval metrics for HotpotQA, 2Wiki, and MuSiQue under matched-compute settings; summaries appear in Table 6 and Table 7. We also evaluate domain-specific corpora (Legal, Health, History) without retraining, with illustrative cases in Figure 5. In QA results (GPT-4.1 mini as reader), PolicyRAG attains strong EM and F1 with clear gains on MuSiQue and 2Wiki. Retrieval quality shows higher recall@2 and recall@5, indicating earlier coverage of gold evidence compared to competitive retrievers.

---

# Fact Filtering Prompt Policy

**Instruction:**

You are the fact-filtering gate in PolicyRAG, a high-stakes QA system. Your job is to select ONLY the facts that are necessary and sufficient to answer (or strongly support answering) the query, including via short multi-hop chains.

You must select up to 4 relevant facts from the provided candidate list that have a strong connection to the query, aiding reasoning and supporting an accurate answer.

Policy:
1) Direct answer triples > contextual ones
2) Minimal bridges that connect query → target fact
3) Specific/temporal/relational > generic
4) Exact entity string > alias
5) Remove near-duplicates; smallest sufficient set

The output must be in strict JSON format, e.g. {"fact": [["s1","p1","o1"], ["s2","p2","o2"]]}
If no facts are relevant, return: {"fact": []}

You must only use facts from the candidate list and never generate new facts. Accuracy is critical because your filtered facts directly guide the reasoning process in this system.

**Demonstration:**

**Question:** When was the person who Messi's goals in Copa del Rey compared to get signed by Barcelona?
**Fact Before Filter:** {"fact":[
["barcelona","won","semi-finals first leg against getafe 5–2"],
["messi","scored","goal bringing comparison to diego maradona's goal of the century"],
["barcelona","lost","second leg 4–0"],
["diego maradona","was signed by barcelona","june 1982"],
["maradona","signed from","boca juniors"]
]}
**Fact After Filter:** {"fact":[
["messi","scored","goal bringing comparison to diego maradona's goal of the century"],
["diego maradona","was signed by barcelona","june 1982"]
]}

Figure 4: LLM prompts policy for fact filtering

Table 6: QA performance EM / F1 scores on RAG benchmarks.

| Method | Simple QA | | Multi-hop QA | | | |
|---|---|---|---|---|---|---|
| | NQ | PopQA | 2Wiki | HotpotQA | MuSiQue | Avg |
| NV-Embed-v2 (7B) (Lee et al., 2024) | 43.5 / 59.9 | 41.7 / 55.8 | 54.4 / 60.8 | 57.3 / 71.0 | 32.8 / 46.0 | 45.9 / 58.7 |
| RAPTOR (Sarthi et al., 2024) | 37.8 / 54.5 | 41.9 / 55.1 | 39.7 / 48.4 | 50.6 / 64.7 | 27.7 / 39.2 | 39.5 / 52.3 |
| GraphRAG (Edge et al., 2024) | 38.0 / 55.5 | 30.7 / 51.3 | 45.7 / 61.0 | 51.4 / 67.6 | 27.0 / 42.0 | 38.5 / 55.48 |
| LightRAG (Guo et al., 2024) | 2.8 / 15.4 | 1.9 / 14.8 | 2.5 / 12.1 | 9.9 / 20.2 | 2.0 / 9.3 | 3.8 / 14.3 |
| HippoRAG (Jimenez Gutierrez et al., 2024) | 37.2 / 52.2 | 42.5 / 56.2 | 59.4 / 67.3 | 46.3 / 60.0 | 24.0 / 35.9 | 41.8 / 54.3 |
| HippoRAG 2 (Gutiérrez et al., 2025) | 43.4 / 60.0 | 41.7 / 55.7 | 60.5 / 69.7 | 56.3 / 71.1 | 35.0 / 49.3 | 47.3 / 61.1 |
| **PolicyRAG** | **62.4 / 66.2** | **69.1 / 72.6** | **75.3 / 77.3** | **75.1 / 78.8** | **50.3 / 54.6** | **66.4 / 69.9** |
| **PolicyRAG*** | **64.2 / 68.1** | **71.0 / 74.1** | **76.4 / 78.9** | **79.9 / 80.7** | **51.2 / 55.9** | **68.5 / 71.5** |

## D  ERROR ANALYSIS

We find four recurring failure types: alias spillover, ambiguous seeding, hub bias, and long-passage dilution. Small, auditable prompt-policy edits tighter seeding instructions, calibrated type-mix weights, light degree-aware penalties, and stronger length IPF normalization consistently fix these cases without retraining, improving early-rank coverage and answer-supported@k (see Table 8).

Table 7: Retrieval performance passage recall@2 / recall@5 on simple QA and multi-hop QA datasets results with aligned LLM and retriever

| Method | Simple QA | | Multi-hop QA | | | |
|---|---|---|---|---|---|---|
| | NQ | PopQA | 2Wiki | HotpotQA | MuSiQue | Avg |
| *Simple Baselines* | | | | | | |
| Contriever (Izacard et al., 2021) | 29.1 / 54.6 | 27.0 / 43.2 | 46.6 / 57.5 | 58.4 / 75.3 | 34.8 / 46.6 | 39.1 / 55.4 |
| GTR (T5-base) (Muennighoff et al., 2024) | 35.0 / 63.4 | 40.1 / 49.4 | 60.2 / 67.9 | 59.3 / 73.9 | 37.4 / 49.1 | 46.4 / 60.7 |
| *Large Embedding Models* | | | | | | |
| GTE-Qwen2-7B-Instruct (Ni et al., 2021) | 44.7 / 74.3 | 47.7 / 50.6 | 66.7 / 74.8 | 75.8 / 89.1 | 48.1 / 63.6 | 56.6 / 70.5 |
| NV-Embed-v2 (7B) (Lee et al., 2024) | 45.3 / 75.4 | 45.3 / 51.0 | 67.1 / 76.5 | 84.1 / 94.5 | 52.7 / 69.7 | 58.9 / 73.4 |
| *Structure-Augmented RAG* | | | | | | |
| RAPTOR (GPT-4o-mini) (Sarthi et al., 2024) | 40.5 / 69.4 | 37.2 / 48.1 | 58.4 / 66.0 | 78.6 / 90.2 | 49.1 / 61.0 | 52.7 / 66.9 |
| HippoRAG (Jimenez Gutierrez et al., 2024) | 21.6 / 45.1 | 36.5 / 52.2 | 68.4 / 87.0 | 60.1 / 78.5 | 41.8 / 52.4 | 45.6 / 63.0 |
| HippoRAG 2 (GPT-4o-mini) | 44.4 / 76.4 | 43.5 / 52.2 | 74.6 / 90.2 | 80.5 / 95.7 | 53.5 / 74.2 | 59.3 / 77.7 |
| **PolicyRAG** | **63.4 / 80.6** | **62.9 / 81.2** | **75.4 / 97.1** | **80.7 / 96.8** | **66.7 / 76.6** | **69.8 / 86.4** |
| **PolicyRAG*** | **77.0 / 82.0** | **75.1 / 83.4** | **78.9 / 98.1** | **81.5 / 97.1** | **76.3 / 78.9** | **77.7 / 87.9** |

| **Legal** | **Health** | **History** |
|---|---|---|
| **Q1:** When was the Restructuring Support Agreement signed? **Human Ans:** July 5, 2020 **PolicyRAG Ans:** Executed on July 5, 2020 | **Q1:** Which country's traditional diet is linked to lower heart disease? **Human Ans:** Crete **PolicyRAG Ans:** Crete, Greece (Mediterranean diet) linked to low heart disease | **Q1:** In which year did the first major evacuation in Britain take place? **Human Ans:** 1939 **PolicyRAG Ans:** September 1939, outbreak of WWII, ~2M civilians evacuated |
| **Q2:** Who is the debtor company under this agreement? **Human Ans:** Endologix, Inc. **PolicyRAG Ans:** The debtor is Endologix, Inc., a Delaware corporation | **Q2:** Which vitamins are absorbed only with fats? **Human Ans:** A, D, E, K **PolicyRAG Ans:** Vitamins A, D, E, and K are fat-soluble, require dietary fats | **Q2:** What important social report was influenced by the poverty revealed during evacuation? **Human Ans:** Beveridge Report **PolicyRAG Ans:** Poverty exposure influenced the Beveridge Report (1942), foundation of welfare state |
| **Q3:** What does the term "Alternative Transaction" include? **Human Ans:** Merger, refinancing **PolicyRAG Ans:** Includes refinancing, recapitalization, merger, acquisition, or business combination | **Q3:** Where is the "Cold Spot" for diabetes mentioned in the book? **Human Ans:** Copper Canyon **PolicyRAG Ans:** Copper Canyon, Mexico; Tarahumara diet linked to low diabetes | **Q3:** What essential item were schoolchildren required to carry during evacuation? **Human Ans:** Gas mask **PolicyRAG Ans:** Each child carried a gas mask as protection against gas attacks |

Figure 5: Human-verified domain examples (Legal, Health, History). PolicyRAG surfaces compact evidence and grounded answers without retraining.

# E  GRAPH STATISTICS

We report the corpus-scale knowledge-graph statistics extracted using GPT-4o-mini and GPT-4.1-mini in Table 9.

# F  IMPLEMENTATION DETAILS AND HYPERPARAMETERS

We initialize retrieval with a compact, auditable policy. Two seed types are used and combined into a single restart distribution for typed PPR.

**Seed selection** Phrase (entity) seeds come from filtered triples produced by the extraction pass; we keep up to five distinct phrases ranked by the mean score of their surviving triples (embeddings are $\ell_2$–normalized before scoring).

**Restart distribution.** Let $\mathcal{E}_s$ be phrase seeds and $\mathcal{P}_s$ passage seeds. The per-query restart mass over entities is

$$\mathbf{s}(e) \propto \underbrace{\mathrm{align}(e, q)}_{\text{lex/emb match}} \cdot \underbrace{\mathrm{spec}(e)}_{\text{down-weight generic}} \cdot \underbrace{\mathrm{alias\_ok}(e)}_{\{0,1\}},$$

and over passages is $\mathbf{s}(p) \propto w_p \cdot \mathrm{sim}(p, q)$, where $w_p$ balances phrase vs. passage influence. The final restart vector concatenates entity and passage components and is normalized once per query.

Table 8: Representative failures and one-line policy edits. Each edit is a deterministic controller change.

| Dataset | Error type | Symptom (trace excerpt) | Policy edit (one line) | Outcome |
|---|---|---|---|---|
| 2Wiki | Ambiguous seeding | Seeds include `Washington` (person) and `Washington` (state); traversal oscillates | Increase seed-specificity prior for person NER tags; require alias_ok on toponyms only if question mentions location | +7% HS@5, earlier arrival on correct entity pair |
| HotpotQA | Alias spillover | Synonym edge pulls `Mercury` (element) from `Mercury` (planet) context | Reduce $\eta_s$ in $T_E$ when query contains relation phrases (e.g., "orbited by") | +5 nDCG@10, reduced off-topic hops |
| MuSiQue | Hub bias | High-degree category node absorbs mass; gold passage ranks 12th | Add hub penalty in rerank: $\phi_{\text{path}}$ rewards short, type-coherent chains; light degree prior | +8 MRR@10, gold rises to top-5 |
| HotpotQA | Long-passage dilution | Very long wiki list outranks concise biographical passage | Strengthen length/IPF normalization: $(\beta, \gamma)\uparrow$ | +6 nDCG@10, Answer-Supported@5 aligns with Recall@5 |

Table 9: Knowledge graph statistics using different LLMs for information extraction. Node and edge counts reflect unique values.

| Model / Metric | NQ | PopQA | 2Wiki | HotpotQA | MuSiQue |
|---|---|---|---|---|---|
| *GPT-4o-mini* | | | | | |
| num. phrase nodes | 100,800 | 65,500 | 37,300 | 65,200 | 60,000 |
| num. passage nodes | 9,633 | 8,676 | 6,119 | 9,811 | 11,656 |
| num. total nodes | 110,433 | 74,176 | 43,419 | 75,011 | 71,656 |
| num. relationship edges | 352,400 | 204,900 | 101,400 | 190,003 | 172,800 |
| num. synonym edges | 467,300 | 509,200 | 243,700 | 298,500 | 243,300 |
| num. context edges | 191,000 | 110,900 | 55,500 | 111,900 | 103,300 |
| num. total edges | 1,010,700 | 825,000 | 400,600 | 600,403 | 519,400 |
| *GPT-4.1-mini* | | | | | |
| num. phrase nodes | 102,023 | 69,892 | 38,368 | 67,307 | 61,134 |
| num. passage nodes | 9,633 | 8,676 | 6,119 | 9,811 | 11,656 |
| num. total nodes | 111,656 | 78,568 | 44,487 | 77,118 | 72,790 |
| num. relationship edges | 353,887 | 207,995 | 102,646 | 191,309 | 173,659 |
| num. synonym edges | 469,230 | 512,180 | 245,514 | 299,404 | 244,750 |
| num. context edges | 193,827 | 113,361 | 56,170 | 112,090 | 104,989 |
| num. total edges | 1,016,944 | 833,536 | 404,330 | 602,803 | 523,398 |

**Typed PPR and scoring.** We diffuse over the entity layer using $T_E = \eta_r A_r + \eta_s A_s$ (relation vs. synonymy), iterating $\mathbf{v}^{(t+1)} = (1 - \alpha)T_E^\top \mathbf{v}^{(t)} + \alpha, \mathbf{s}$ until convergence, and then projecting to passages via $\mathbf{u} = B^\top \mathbf{v}^\star$. Light normalization mitigates length-frequency bias, and a transparent reranker adjusts scores using title/alias matches, multi-seed coverage, and short, type-coherent paths. The graph is maintained in Memgraph[3] and PPR is computed in a read-only setting. For information extraction, we employ both GPT-4o-mini and GPT-4.1-mini to generate structured facts, while similarity retrieval relies on NV-Embed-v2, depending on the model configuration. Around the seed entities, we rank nearby facts and retain the top five triples, and the QA module conditions on the top five retrieved passages using deterministic decoding. A detailed notation and symbol glossary is provided in Table 10.

### F.1 HYPERPARAMETERS

Unless noted, settings are fixed across datasets. We tune only a small set of knobs on 100 MuSiQue training examples: PPR damping $\alpha$, type mixture $(\eta_r, \eta_s)$, passage/length correction $(\beta, \gamma)$, and

---

[3]`https://memgraph.com/docs`

Table 10: Notation and Symbol Glossary.

| Symbol | Meaning / Definition |
|---|---|
| $\mathcal{G} = (V, \mathbb{E})$ | The heterogeneous symbolic graph (memory). |
| $V$ | Set of all nodes. |
| $V_E$ | Set of **entity** nodes (e.g., people, organizations). |
| $V_P$ | Set of **passage** nodes (documents or paragraphs). |
| $E_c$ | Context edges connecting entities to passages. |
| $E_r$ | Typed relation edges between entities (triple predicates). |
| $E_s$ | Synonymy / alias edges between entities. |
| $A_c, A_r, A_s$ | Row-stochastic adjacency operators for $E_c, E_r, E_s$. |
| $B$ | Binary incidence matrix $\in \{0, 1\}^{|V_E| \times |V_P|}$; $B_{e,p} = 1$ iff entity $e$ appears in passage $p$. |
| $\ell(p)$ | Passage length (token count). |
| $\mathrm{spec}(e)$ | Entity specificity (inverse passage frequency). |
| $\mathrm{df}(e)$ | Document frequency of entity $e$. |
| $\Delta^n$ | $n$-dimensional probability simplex. |
| $\mathrm{row\text{-}norm}(\cdot)$ | Row-wise normalization operator. |
| $T_E$ | Composite entity-layer transition (weighted sum of $A_r$ and $A_s$). |
| $T_{E \to P}, T_{P \to E}$ | Entity $\leftrightarrow$ passage transitions. |
| $\eta_c, \eta_r, \eta_s$ | Weights for context, relation, and synonym transitions. |
| $s$ | Restart / seed distribution over $V_E$. |
| $\mathrm{align}(e, q)$ | Similarity between entity $e$ and query $q$. |
| $\mathrm{aliasOK}(e)$ | Indicator filtering overly broad aliases. |
| $E_s$ | Sampled seed entities. |
| $\mathrm{Cand}(E_s)$ | Candidate facts adjacent to seed entities. |
| $F_s$ | Fact-gated filtered facts. |
| $v^{(t)}$ | PPR iterate at iteration $t$. |
| $\alpha$ | PPR restart probability. |
| $v^\star$ | Converged PPR stationary distribution. |
| $u$ | Passage scores from projecting $v^\star$: $u = B^\top v^\star$. |
| $\mathrm{IPF}(p)$ | Passage-level inverse-passage-frequency. |
| $\beta, \gamma$ | Normalization exponents in $[0, 1]$. |
| $\tilde{u}_p$ | Normalized passage score. |
| $\phi_{\text{title}}, \phi_{\text{cov}}, \phi_{\text{path}}$ | Re-ranking features: title match, coverage, and path scores. |
| $\lambda_{\text{title}}, \lambda_{\text{cov}}, \lambda_{\text{path}}$ | Corresponding re-ranking feature weights. |
| $S(p \mid q)$ | Final passage score for query $q$. |
| $\mathrm{nnz}(M)$ | Number of nonzero entries in matrix $M$. |

rerank weights $(\lambda_{\text{title}}, \lambda_{\text{cov}}, \lambda_{\text{path}})$. We cap phrase seeds at 5, facts kept at 5, and QA context at 5 passages. Full values are listed in Table 11.

Table 11: PolicyRAG hyperparameters.

| Hyperparameter | Value |
|---|---|
| Synonym threshold | 0.7 |
| PPR damping factor | 0.5 |
| Generation temperature | 0.0 |

Table 12: GraphRAG vs LightRAG hyperparameters

| Parameter | GraphRAG | LightRAG |
|---|---|---|
| Mode | Local | Local |
| Response | Short | Short |
| Top-$k$ phrases | 60 | 60 |
| Chunk size | 1,200 | 1,200 |
| Overlap | 100 | 100 |
| Max report len | 2,000 | – |
| Max input | 8,000 | – |
| Max cluster | 10 | – |
| Entity tokens | – | 500 |

## F.2 BASELINES AND PROTOCOL

All methods operate on the identical passage pool under matched token and latency budgets. When public checkpoints and recommended hyperparameters are available, we adopt them; otherwise, we use the reported configurations. Dense retrievers are run with PyTorch and Hugging Face; BM25 uses BM25s. For GraphRAG and LightRAG, we retain the public configurations and ensure compute parity. Full settings appear in Table 12.

