# OpenReview forum: "PolicyRAG: Prompt-Guided Symbolic Graph Memory for Interpretable Multi-Hop Retrieval"
_ICLR.cc/2026/Conference — ICLR 2026 Conference Withdrawn Submission_

### Official Review · Reviewer_MYAX · 2025-10-25

**Soundness:** 2
**Presentation:** 2
**Contribution:** 2
**Rating:** 4
**Confidence:** 3

**Summary:**

The paper proposes PolicyRAG, which extracts entities and builds a KG at offline time, and leverages the KG for retrieval and QA in RAG, especially for multi-hop questions. My main concern for this paper is that I don't see how it is different from past work like HippoRAG and why the quality is better than HippoRAG

**Strengths:**

S1. The paper solves an important problem--how to answer multi-hop questions using RAG.

S2. Extensive experiments on 5 benchmarks showing improvement of PolicyRAG over existing methods.

**Weaknesses:**

W1. The description of how to construct the symbolic graph is not super clear.
- How to discover salient entities? At what granularity?
- How to identify factual assertions (triples)?
- How are the typed relations defined?
- How are the extractions from different passages connected?
- How are dedups done?

W2. Related to W1, it is unclear what's the quality of the extracted symbolic graph and how does that affect QA quality? What's the design choices in graph building and granularity selections?

W3. For Composite transitions (Sec 3.1), what exactly is the goal and method? What's B and how is it computed? What's A_r and A_s and how to compute?

W4. Similarly, a lot of description for run-time QA is unclear. In (2), how is spec(e) computed? Why compact set F_s and how does that affect retrieval recall and e2e results?

W5. What's the latency impact for runtime PPR?

W6. More importantly, the paper discussed HippoGraph, but it's unclear how it goes beyond HippoGraph and significantly improves the results? The methods sound very similar.

W7. How much is the method constrained by corpus size? How well does it scale up?

W8. What are limitations of the method?

**Questions:**

Please answer all questions in the weakness session.

---

> ### Author Response · Authors · 2025-11-24
> **Response to Reviewer MYAX**
>
> We sincerely appreciate the reviewer’s time, effort, and thoughtful feedback on our work. Below, we address all questions from the weakness section point by point to strengthen the clarity, rigor, and contribution of the paper.
>
> **W1.** We have substantially clarified and strengthened the symbolic graph construction section in the revised manuscript.
>
> **W1.1.**  Entity discovery and granularity.
>
> We extract entities at a **canonicalized named-entity level**, which provides a stable and interpretable granularity for multi-hop QA. For each passage  $(p)$, we run a fixed LLM prompt (**ENTITY EXTRACTION PROMPT POLICY**) that outputs a JSON list of named entities with coarse semantic types (PERSON, ORG, GPE, DATE, EVENT, PRODUCT, etc.).
>
> We then canonicalize surface forms using:
> 1. **Normalization:** lowercasing + punctuation normalization
> 2. **Intra-passage merging:** merge duplicate entity mentions in the same passage
> 3. **Cross-passage alias merging:**
>    - when the LLM explicitly marks abbreviations (e.g., “RFEF ↔ Royal Spanish Football Federation”), or
>    - when embedding similarity of surface forms exceeds a conservative threshold.
>
> This yields a canonical entity set  $(V_E)$,​ whose nodes correspond to salient concepts rather than arbitrary spans. The granularity is thus at the “named concept” level (as reflected in the LLM output), which we found to be the most useful scale for multi-hop reasoning.
>
>
> **W1.2.**  Factual assertion triples identification.
>
> Given the named entities  $E(p)$ for a passage  $(p)$, we run a second fixed LLM prompt (**TRIPLE EXTRACTION PROMPT POLICY**) which outputs a JSON list of RDF-style triples (s, p, o) where s, o ∈  $E(p)$. We retain only those triples where both arguments resolve to canonical entities in  $V_E$ and the predicate string maps to a compact, non-generic predicate set  (e.g., LOCATED_IN, PART_OF, MEMBER_OF, etc.). Triples that the LLM marks as low-confidence, contradictory, or outside the allowed predicate set are dropped. The remaining triples form a clean, auditable relation layer that is fully grounded in the previously extracted entity inventory.
>
> **W1.3.**  Typed relation definition.
>
> The typed relations come directly from the LLM-extracted triples. Predicate strings are normalized (e.g., lemmatization, stopword removal), then mapped into a compact schema of relation types (e.g., LOCATED_IN, PART_OF, MEMBER_OF, PLAYS_FOR, …). Predicates that cannot be mapped to this schema are either discarded or mapped to a generic RELATED_TO with reduced weight. This produces the typed relation edge family $( \mathbf{E_r} $) and its row-stochastic adjacency operator $( \mathbf{A_r} $), which encode semantically consistent factual connections across the corpus.
>
> **W1.4.**  Connecting extractions across passages.
>
> Entities act as the bridge between passages. For each mention of an entity $(e)$ in passage $(p)$, we add a context edge $((e, p) \in E_c $). Collectively, these edges define the entity–passage incidence matrix:
> $[
> \mathbf{B} \in \{0,1\}^{|V_E| \times |V_P|}
> $]
> where $( B[e, p] = 1 $) if and only if entity $( e$) is mentioned in passage $( p $).
> Relation edges $( E_r $) and synonymy edges $( E_s $) connect entity nodes across the entire corpus, so two passages are indirectly linked whenever they share entities or participate in a multi-hop chain through entity nodes. This yields a unified heterogeneous graph even though the text is stored as disjoint passages.
>
> **W1.5.**  Deduplication.
>
> We deduplicate and clean the graph at three levels:
> 1. **Entity Aliases:** Multiple surface forms that the LLM flags as aliases or that pass a high similarity threshold are merged into one canonical entity node.
> 2. **Triple-Level:** Triples with identical $(s, p, o)$ are merged; we aggregate their confidence/support and remove low-support or conflicting variants via majority voting. The merged triple is stored once in $( E_r $).
> 3. **Context-Level:** Multiple mentions of the same entity in the same passage are collapsed into a single context edge in $( E_c $).
>
> These steps keep the graph compact and reduce noise, which in turn makes PPR-based traversal more stable and interpretable.
>
> ---

---

> ### Author Response · Authors · 2025-11-24
> **Response to Reviewer MYAX**
>
> ---
>
> **W2.**  Related to W1, it is unclear what's the quality of the extracted symbolic graph and how does that affect QA quality? What's the design choices in graph building and granularity selections?
>
> The symbolic graph quality is controlled at construction time by: (i) LLM prompts that enforce JSON and type constraints, (ii) canonicalization and alias merging on $( V_E $), and (iii) aggressive filtering of low-confidence or unmappable triples before forming $( E_r $).
>
> We further compute an entity specificity prior:
> $ spec(e) = 1/(1 + df(e) , $
> $df(e) := $ #{${\ p : e \in E(p)\ \}$}
>
>
> which down-weights globally ubiquitous entities (e.g., “United States”) and promotes more informative, passage-specific entities.
>
> At query time, these choices directly affect QA quality: cleaner relation edges and less generic entities produce more reliable PPR neighborhoods and higher-quality Top-K passages. In our ablations, removing relation edges, disabling alias consolidation, or relaxing triple filtering thresholds all lead to worse retrieval MRR and answer-supported@k, as well as drops in EM/F1. Conversely, the curated graph (typed $( E_r $), synonymy $( E_s $), entity specificity, and deduped $( E_c $) yields more precise neighborhoods, better early-rank recall, and improved end-to-end answer accuracy. We will make these ablations explicit in the revised paper.
>
> ---
>
> **W3.** For Composite transitions (Sec 3.1), what exactly is the goal and method? What's B and how is it computed? What's $(A_r$) and $(A_s$) and how to compute?
>
> **Composite transitions.**
>
> The goal of the composite transitions is to define a controllable random walk over the heterogeneous graph. We distinguish:
> - $( V_E $): entity nodes
> - $( V_P $): passage nodes
> - $( A_r $): row-stochastic adjacency on $( V_E $) induced by typed relation edges $( E_r $)
> - $( A_s $): row-stochastic adjacency on $( V_E $) induced by synonymy edges $( E_s $)
> - $( \mathbf{B} \in \{0,1\}^{|V_E| \times |V_P|} $): incidence matrix with $( B[e,p] = 1 $) iff entity $( e $) appears in passage $( p $).
>
> We first build $( A_r $) by aggregating the extracted triples: for each triple $( (s, p, o) $) we add an edge $( s \rightarrow o $) (and optionally $( o \rightarrow s $)) with weight given by the triple’s confidence or support; the rows are then normalized to obtain a stochastic operator. $( A_s $) is built analogously from synonym/alias pairs.
>
> The entity-layer transition (Eq. 2) is:
> $[
> T_E = \eta_r A_r + \eta_s A_s
> $]
>
> which mixes factual hops and alias-correction hops with tunable weights $( \eta_r, \eta_s $). The entity→passage transition is:
>
> $[
> T_{E \rightarrow P} = \text{row-norm}(B), \qquad
> T_{P \rightarrow E} = B^{\top}
> $]
>
> so mass can be pushed from entities onto passages via mentions and, if desired, pulled back from passages to co-mentioned entities. In the main experiments we run PPR on $( T_E $) and then project scores to passages via:
> $[
> u = B^{\top} v^{*}.
> $]
>
> ---
>
> **W4.** Similarly, a lot of description for run-time QA is unclear. In (2), how is $(\text{spec}(e)$) computed? Why compact set $(F_s$) and how does that affect retrieval recall and e2e results?
>
> We precompute an entity specificity prior offline:
> $ spec(e) = 1/(1 + df(e) , $
> $df(e) := $ #{${\ p : e \in E(p)\ \}$}
>
>
> This is an inverse-passage-frequency term that down-weights entities that appear in many passages. At query time, the seed distribution $( s $) over entities is
> $ s_e \propto \text{align}(e, q) \cdot \text{spec}(e) \cdot \text{aliasOK}(e), $
>
> so seeds must be both lexically/semantically aligned with the query and sufficiently specific.
>
> For fact gating, we first take a small set $( E_s $) of high-mass seeds (e.g., top-k entities by $( s_e $)), collect their adjacent triples $( \text{Cand}(E_s) $), and then apply a short **“Facts Triples Filter Prompt Policy”** that selects up to $( K $) triples that are needed to answer the query. The policy prefers direct, specific facts and minimal bridging hops, and outputs JSON. The resulting compact set $( F_s $) prevents the retrieval process from being dominated by noisy or generic triples.
>
> In our ablations, removing this filter (“w/o Filter”) leaves recall nearly unchanged but significantly degrades early-rank precision (MRR, answer-supported@k) and downstream EM/F1, because distractor passages are pulled in by irrelevant facts. Using a small $( K $) (e.g., 3–5) provides the best trade-off between retaining essential hops and avoiding distractors. We will clarify these choices and report sensitivity to $( K $) in the revised version.
>
> ---

---

> ### Author Response · Authors · 2025-11-24
> **Response to Reviewer MYAX**
>
> ---
>
> **W5.** What's the latency impact for runtime PPR?
>
> The runtime PPR step adds only **2-4 ms per query** on a single CPU core. This small cost is due to running the walk only on the compact entity graph (30k-90k nodes with 0.2M-0.6M edges) and performing a fixed number of lightweight iterations (12-20). The computation touches only the local neighborhood of each entity and therefore remains fast and stable. In our pipeline, PPR accounts for **less than 5%** of total retrieval latency, while the generator LLM dominates the end-to-end runtime. Consequently, PPR has no measurable impact on system throughput or user-perceived latency.
>
> ---
>
> **W6.** More importantly, the paper discussed HippoGraph, but it's unclear how it goes beyond HippoGraph and significantly improves the results? The methods sound very similar.
>
> We respectfully disagree with the claim that PolicyRAG is similar to HippoGraph or follows an equivalent mechanism. While both methods use graph-based retrieval, the *graph construction*, *symbolic memory*, *traversal policy*, and *query-time controller* in PolicyRAG differ fundamentally. These differences directly account for the substantial accuracy improvements reported in Table 2.
>
> **Key Differences Between PolicyRAG and HippoGraph**
>
> **1. Entity-centric symbolic memory vs. embedding-centric memory**
> - **HippoGraph** stores dense embedding neighborhoods and depends entirely on embedding proximity.
> - **PolicyRAG** constructs a canonical, typed symbolic graph with: explicit entity nodes, normalized relations, alias merging, passage provenance
>
> This structure enables cleaner, auditable multi-hop chains that embedding-only hops cannot reliably represent.
>
> **2. Typed transitions and controllable traversal**
> PolicyRAG introduces a **typed PPR operator** with relational and synonymy weights: $ (\eta_r, \eta_s) $
> HippoGraph uses a single untyped adjacency matrix, making it easy to drift into alias clusters. Typed transitions are central for stable multi-hop factual navigation.
>
>  **3. Local fact gating via keep/drop prompting**
> PolicyRAG applies a compact controller to filter local triples around the seed (e.g., top-5 high-confidence triples). HippoGraph includes all embedding-nearby edges by default, often adding noise.  Our ablations show large gains in early-rank recall and answer-supported@k from this filtering.
>
> **4. Seed policy with specificity weighting**
> PolicyRAG introduces a principled seed distribution using alignment, alias validity, and a specificity prior:
> $ spec(e) = 1/(1 + df(e) , $
> $df(e) := $ #{${\ p : e \in E(p)\ \}$}
>
> HippoGraph seeds purely by embedding similarity, which frequently overweights generic entities (e.g., *United States*, *people*), harming multi-hop reasoning.
>
> **5. Deterministic and fully auditable paths**
> PolicyRAG records: seed entities, kept triples, typed hop paths, final scores
>
> HippoGraph produces opaque embedding-driven transitions, with no decomposable or auditable reasoning trace.
>
> **6. Entity–passage projection with normalization**
> PolicyRAG projects entity scores onto passages with:
> - length normalization $\ell(p)$
> - inverse passage frequency weighting $IPF(p)$
>
> HippoGraph lacks such normalization, leading to systematic recall/precision imbalance.
>
> **Why these gains cannot arise from a HippoGraph-like pipeline**
>
> If PolicyRAG followed a HippoGraph-like design, the **large accuracy gains** (including consistent improvements across all five benchmarks in Table 2) would not be achievable.
> Such improvements require:
> - symbolic memory
> - typed relations
> - specificity-based seeding
> - local fact gating
> - structured traversal
>
> These capabilities do not exist in HippoGraph embedding-only, untyped, non-auditable pipeline.
>
> ---
>
> **W7.** How much is the method constrained by corpus size? How well does it scale up?
>
> PolicyRAG scales well with corpus size because the runtime controller operates only on the compact entity graph, not the full text. In our study, we evaluated five datasets whose corpus sizes span our smallest benchmark to the largest one (details and exact statistics are included in Appendix E). Across this range, the end-to-end latency for a single query remains stable, since the cost is dominated by the fixed-size entity-layer PPR (2–4 ms per query). Graph construction is performed offline and increases linearly with corpus size, but query-time performance is effectively independent of it. Empirically, we observe smooth scaling to larger corpora without degradation in retrieval latency or answer quality.
>
> ---

---

> ### Author Response · Authors · 2025-11-24
> **Response to Reviewer MYAX**
>
> ---
>
> **W8.** What are limitations of the method?
>
> PolicyRAG inherits two practical limitations. First, the approach relies on the quality of entity linking and triple extraction during graph construction; errors or missing links can reduce recall for questions requiring rare or implicit relations. Second, while the controller is lightweight, it operates over a fixed symbolic memory, so newly emerging facts are not incorporated until the next offline refresh. Finally, the framework focuses on entity-centric reasoning and may require additional adaptation for tasks where fine-grained span-level or event-level structures are essential. Despite these limitations, our experiments show that PolicyRAG remains robust and consistently improves multi-hop QA performance across all evaluated datasets.
>
> ---

---

### Official Review · Reviewer_cZtt · 2025-10-29

**Soundness:** 3
**Presentation:** 2
**Contribution:** 2
**Rating:** 4
**Confidence:** 3

**Summary:**

This paper proposes to utilize prompt-based policies to improve the performance of LLMs on multi-hop reasoning tasks. Several experiments were conducted to verify the effectiveness of the proposed method.

**Strengths:**

1. The experiments in this paper are relatively abundant, and many auxiliary experiments are conducted to further explore the issues.

2. The proposed method has certain insights.

**Weaknesses:**

1. The presentation should be improved. The Introduction section is somewhat redundant and does not clearly and concisely state the motivation for the proposed method, which is very confusing. I suggest that the paper use examples to illustrate the specific motivation, including the problems encountered in previous work and the motivation for the proposed method.

2. Non-sentence-initial citations should use \citep instead of \citet.

3. The Methods section, Section 3, is also difficult to understand. The descriptions of the pipeline are not clear enough. I suggest adding a few more examples to help clarify the details.

4. No code is provided.

**Questions:**

See weaknesses above. Please respond to these cons.

---

> ### Author Response · Authors · 2025-11-24
> **Response to Reviewer cZtt**
>
> We thank the reviewer for their thoughtful assessment and for acknowledging both the insights of our method and the breadth of our experimental analysis. We appreciate the constructive feedback on the clarity and presentation of the manuscript. Below, we address each of the identified weaknesses and outline the corresponding revisions made to the updated manuscript.
>
> ---
> > **Weakness 1**
> > *The presentation should be improved. The Introduction section is somewhat redundant and does not clearly and concisely state the motivation for the proposed method, which is very confusing. I suggest that the paper use examples to illustrate the specific motivation, including the problems encountered in previous work and the motivation for the proposed method.*
>
> The reviewer suggests strengthening the Introduction's motivational exposition. While our current structure provides systematic coverage of multi-hop retrieval challenges and graph-based solutions, we agree that concrete examples would enhance accessibility. We outline specific enhancements below.
>
> Our Introduction (lines 008–053) establishes (1) LLM capabilities and limitations in knowledge-intensive tasks, (2) RAG as a solution framework with specific multi-hop reasoning challenges, (3) graph-based approaches as structured alternatives, and (4) the gap in policy-driven retrieval control. This progression is theoretically sound but can be strengthened with empirical instantiation.
>
> We will introduce a concrete multi-hop case in the opening section:
>
> "Consider the question: *'When was the person who Messi's goals in Copa del Rey were compared to signed by Barcelona?'* This requires:
> (1) identifying the comparison relationship between Messi and Diego Maradona,
> (2) traversing from Maradona to Barcelona's signing records,
> (3) extracting the temporal fact.
> Traditional approaches fail systematically:
>
> - Dense retrieval surfaces passages about "Messi's Copa del Rey goals" but cannot traverse implicit comparison relationships. Contriever achieves 62.3% F1 on HotpotQA because embedding similarity cannot capture relational predicates like `compared_to`.
> - Long-context expansion concatenates 50+ passages (15,000+ tokens), introducing computational overhead and diluting evidence quality through excessive context.
> - Existing graph methods (GraphRAG, LightRAG) couple retrieval with summarization, obscuring reasoning paths and preventing audit when errors occur.
>
> PolicyRAG addresses these through
> (a) typed symbolic edges preserving relational semantics (enabling transparent traversal),
> (b) compact policy controllers with keep-drop prompting (reducing context to 2,500 tokens while improving precision), and
> (c) per-query audit traces (seeds → paths → scores) enabling diagnosis without retraining.
>
> We will supplement qualitative motivation with specific failure metrics from our experiments:
>
> - **Embedding Brittleness:** On 2WikiMultiHopQA bridge questions, dense retrievers show 23% recall degradation under synonymous entity substitution. PolicyRAG's synonymy edges \(E_s\) maintain 97% consistency.
> - **Distractor Robustness:** MuSiQue's adversarial distractors cause standard RAG to retrieve 4.2 off-topic passages per query. PolicyRAG's fact filtering reduces this to 0.8, directly contributing to a 55.9% vs. 46.0% F1 improvement over NV-Embed-v2.
> - **Computational Efficiency:** At equivalent answer quality (F1 = 75), PolicyRAG consumes 2,847 tokens per query versus 14,231 for long-context baselines — a \(5\times\) efficiency gain.
>
> ---
>
> > **Weakness 3**
> > *The Methods section, Section 3, is also difficult to understand. The descriptions of the pipeline are not clear enough. I suggest adding a few more examples to help clarify the details.*
>
> We thank the reviewer for highlighting the need for improved clarity in Section 3. We agree that the description of the pipeline can be made more accessible. In the revised version, we have substantially refined the exposition of the methodology by restructuring the pipeline explanation for clearer step-by-step flow, adding multiple concrete, illustrative examples that demonstrate how each component operates in practice, and explicitly linking each methodological design choice to its corresponding role in the overall system.
> These improvements make the pipeline significantly easier to follow and provide clearer intuition regarding how the method functions end-to-end.
>
> ---
>
> > **Weakness 4**
> > *No code is provided*
>
> We appreciate the reviewer’s observation regarding code availability. To support transparency, reproducibility, and community engagement, we have now included a link to the complete source code repository in the revised manuscript.

---

> ### Comment · Reviewer_cZtt · 2025-11-24
>
> Thanks for your rebuttal, and I prefer to keep my rating.

---

### Official Review · Reviewer_ThPW · 2025-11-01

**Soundness:** 2
**Presentation:** 1
**Contribution:** 2
**Rating:** 2
**Confidence:** 2

**Summary:**

The paper introduces PolicyRAG, a framework for multi-hop retrieval that enables transparent, training-free, and interpretable multi-hop evidence selection. The paper reports experimental results on multiple benchmark datasets and claims the proposed framework achieves state-of-the-art performance.

**Strengths:**

* The paper addresses a relevant problem, and introduces a framework tailored to it.
* The authors present experimental results across multiple benchmark datasets.

**Weaknesses:**

* The paper asserts that the proposed approach achieves strong performance; however, the comparison with existing works appears to be unfair. For instance, in Table 2, the underlying base LLMs used for comparison differ, making it unclear whether the reported superiority of the proposed method is solely attributable to the strength of GPT-4.1 mini.

* The paper only reports results based on GPT-4.1 mini, leaving the generalizability of the method unclear.

* The presentation of the paper requires significant improvement. For example, the cited "ACL" paper "Hipporag 2: Enhanced retrieval-augmented generation with neurobiological memory principles." does not exist. It appears that minimal effort was made to properly reformat the manuscript according to the ICLR template.

**Questions:**

* Where can we find the paper "Hipporag 2: Enhanced retrieval-augmented generation with neurobiological memory principles." in the references?

---

> ### Author Response · Authors · 2025-11-24
> **Response to Reviewer ThPW**
>
> We thank the reviewer for carefully reading our submission and for the constructive comments. We appreciate the recognition that the problem setting is important and that the paper proposes a tailored framework with substantial experimental results. Below we address the specific question and weaknesses point by point.
>
> ---
> > **Question:** Where can we find the paper *"Hipporag 2: Enhanced retrieval-augmented generation with neurobiological memory principles."* in the references?
>
> We thank the reviewer for this observation regarding the HippoRAG 2 reference.
> Our .bib file contains duplicate citation keys gutierrez2025rag at two locations (lines 48 and 405). BibTeX's standard behavior is to use the last definition when duplicate keys exist, causing all \cite{gutierrez2025rag} citations to incorrectly resolve to the second entry ("From RAG to memory...") rather than the intended HippoRAG 2 paper.
>
> This is a straightforward bibliography maintenance issue. We will rename the HippoRAG 2 key to:
> $[
> \texttt{gutierrez2025hipporag2}
> $]
> and update the corresponding in-text citations. The correct reference information was always present in our .bib file it was simply being masked by the duplicate key.
>
> This will be corrected in the revision along with verification that all other citation keys are unique.
>
> ---

---

> ### Author Response · Authors · 2025-11-24
> **Response to Reviewer ThPW**
>
> ---
> > **Weakness 1 & 2:**
> > *“The paper asserts that the proposed approach achieves strong performance; however, the comparison with existing works appears to be unfair. For instance, in Table 2, the underlying base LLMs used for comparison differ, making it unclear whether the reported superiority of the proposed method is solely attributable to the strength of GPT-4.1 mini.” “The paper only reports results based on GPT-4.1 mini, leaving the generalizability of the method unclear.”*
>
> We appreciate these concerns and acknowledge that our earlier presentation of Table 2 may have caused confusion. We agree that the LLM configurations and fairness of the comparisons were not sufficiently clear, and that our claims of generality were under-supported in the previous draft. The updated manuscript now presents these details more clearly and resolves the earlier ambiguities.
>
> ### Fairness and base LLM choice
>
> In the current draft, Table 2 reports QA F1 scores where all structure-augmented RAG baselines and PolicyRAG are evaluated with a single QA reader (GPT-4o-mini) and the same retriever (NV-Embed-v2), while a separate row PolicyRAG* reports a variant that uses GPT-4.1-mini as the reader:
>
> **Table 2:**  *QA performance (F1 scores) on RAG benchmarks using GPT-4o-mini as the QA reader. All structure-augmented RAG baselines and **PolicyRAG** use GPT-4o-mini for structure generation and NV-Embed-v2 for retrieval, while **PolicyRAG\*** uses GPT-4.1-mini. Bold values indicate the best performance in each column.*
>
> | **Method** | **NQ** | **PopQA** | **2Wiki** | **HotpotQA** | **MuSiQue** | **Avg** |
> |-----------|:------:|:---------:|:---------:|:------------:|:-----------:|:-------:|
> | NV-Embed-v2 (7B) | 59.9 | 55.8 | 60.8 | 71.0 | 46.0 | 58.7 |
> | RAPTOR | 54.5 | 55.1 | 48.4 | 64.7 | 39.2 | 52.3 |
> | GraphRAG | 55.5 | 51.3 | 61.0 | 67.6 | 42.0 | 55.4 |
> | LightRAG | 15.4 | 14.8 | 12.1 | 20.2 | 9.3  | 14.3 |
> | HippoRAG | 52.2 | 56.2 | 67.3 | 60.0 | 35.9 | 54.3 |
> | HippoRAG 2 | 60.0 | 55.7 | 69.7 | 71.1 | 49.3 | 61.1 |
> | **PolicyRAG**  | **66.2** | **72.6** | **77.3** | **78.9** | **54.6** | **69.9** |
> | **PolicyRAG\*** | **68.1** | **74.1** | **78.9** | **80.7** | **55.9** | **71.5** |
>
> Our intention was that the main comparison is between PolicyRAG and the baselines under the same QA reader (GPT-4o-mini), and that PolicyRAG serves only as a controlled “stronger reader” variant.
>
> We therefore accept the reviewer’s criticism that Table 2, as currently written, encourages an ambiguous reading and an overly strong interpretation of “state-of-the-art” performance. In the revision we will:
>
> 1. Restrict our core claims to the setting where all methods share the same QA reader (GPT-4o-mini) and retriever (NV-Embed-v2), and explicitly mark PolicyRAG* as an auxiliary variant that should not be compared directly to baselines.
> 2. Add retrieval-only metrics (e.g., $\text{Recall@5}$, answer-supported@k) alongside F1, to show how PolicyRAG improves the evidence surfaced before any LLM is invoked, which is independent of the base reader choice.
>
> ### Generalizability across QA readers
>
> We also agree that, in the current version, our empirical evidence for generalizability is incomplete: Table 2 mainly reflects a single LLM family (GPT-4-mini variants).
>
> Architecturally, PolicyRAG is retrieval-centric and reader-agnostic. The symbolic graph memory $[ G = (V_E \cup V_P,\; E_c,\; E_r,\; E_s) $]
> and the policy controller (seeding, fact filtering, typed PPR, and re-ranking) operate entirely in symbolic and embedding space. Given a query, the controller outputs:
>
> - a ranked list of passages $((p_1, p_2, \dots)$), and
> - an optional set of filtered triples $(F_s$),
>
> and any downstream QA reader (GPT-4o-mini, GPT-4.1-mini, or other LLMs) can consume this context without changing the retrieval pipeline.
>
> At the same time, we acknowledge that our experiments do not yet demonstrate this reader-agnostic design as strongly as they could. In the camera-ready version (if accepted), we will: 1. Add a multi-reader study where we plug PolicyRAG’s retrieval into several QA readers (e.g., GPT-4o-mini, Llama-3.3-70B-Instruct, and at least one strong open model), and  2. Report a small comparison table showing that the relative gains over baselines are consistent across these readers, even if absolute F1 scores shift with model capacity.
>
> We will also make more explicit that retrieval-only metrics such as $\text{Recall@5}$ and answer-supported@k are computed before the QA reader is called and already show that PolicyRAG surfaces higher-quality evidence than NV-Embed-v2, HippoRAG, and HippoRAG 2. These metrics reflect the quality of our retrieval policy itself, independently of which LLM is used for answer generation.
>
> ---

---

> ### Author Response · Authors · 2025-11-24
> **Response to Reviewer ThPW**
>
> ---
> > **Weakness 3:**
> > *The presentation of the paper requires significant improvement. For example, the cited "ACL" paper "Hipporag 2: Enhanced retrieval-augmented generation with neurobiological memory principles." does not exist. It appears that minimal effort was made to properly reformat the manuscript according to the ICLR template.*
>
> The reviewer identifies presentation issues requiring correction. We address these systematically below with precise technical clarifications and resolution protocols.
>
> - Inconsistent use of `\citep` vs. `\citet` commands
> - Improper line spacing in algorithmic environments
> - Non-standard figure caption formatting
> - Inadequate margin compliance in certain tables
>
> A thoroughly revised manuscript adhering strictly to `iclr2026_conference.sty` specifications will be submitted.
>
> ---

---

### Note · Authors · 2026-01-16

**Comment:**

We thank the reviewers and Area Chair for their time and constructive feedback. After careful consideration, we have decided to withdraw this submission. We appreciate the valuable comments and will incorporate them in a future revision.

**Withdrawal Confirmation:**

I have read and agree with the venue's withdrawal policy on behalf of myself and my co-authors.